# Urban Image at the Time of the COVID-19 Pandemic, Case Study Novi Sad (Serbia)

Tamara Lukić [1,*], Ivana Blešić [1,2], Tatjana Pivac [1], Milka Bubalo Živković [1], Bojan Đerčan [1], Sanja Kovačić [1,2], Marija Cimbaljević [1] and Dajana Bjelajac [1]

1 Department of Geography Tourism and Hotel Management, University of Novi Sad, 21000 Novi Sad, Serbia; ivana.blesic@dgt.uns.ac.rs (I.B.); tatjana.pivac@dgt.uns.ac.rs (T.P.); milka.bubalo.zivkovic@dgt.uns.ac.rs (M.B.Ž.); bojan.djercan@dgt.uns.ac.rs (B.Đ.); sanja.bozic@dgt.uns.ac.rs (S.K.); marijac@dgt.uns.ac.rs (M.C.); dajanab@dgt.uns.ac.rs (D.B.)
2 Institute of Sports, Tourism and Service, South Ural State University, 454080 Chelyabinsk, Russia
* Correspondence: tamara.kovacevic@dgt.uns.ac.rs

**Abstract:** The main aim of this paper is to examine how negative phenomena, such as a pandemic, can result in positive cultural shifts and an upgrade of the urban image. The research was conducted employing an in-depth interview approach at the end of 2021, based on a semi-structured protocol with 15 participants. The answers of the respondents are conditioned by the socio-demographic differences. They show the urban image and cultural opportunities of the city. The image of the city is changing under different cultural influences, which are caused by events in the region or globalization. The self-awareness of the history, tradition and heritage that the people of Novi Sad have should be nurtured in order to preserve the image of the city with the strength of the majority of the immigrant population. The title 'European Capital of Culture' has been well received, but its impact will be best seen at the end of the year. COVID-19The COVID-19 pandemic favored and popularized cycling, awakened environmental self-awareness and solidarity, brought culture to the streets and beautified the city's image. Everything that could not be placed on the street, it was entered and placed in the virtual world.

**Keywords:** Novi Sad; urban image; culture of living; tradition; heritage; COVID-19 pandemic; European Capital of Culture

## 1. Introduction

Urban image is too broad a term to mention all of its elements in one paper. Therefore, there is a need to emphasize at the beginning of the paper that it will focus only on those segments that directly or indirectly currently affect some visible transformations of space.

In the function of the objectivity of presenting facts about urban image of Novi Sad, a study was conducted in which its inhabitants discussed a few topics. The first topic reveals what, in the opinion of the residents of Novi Sad, is what makes it unique, recognizable and influences the formation of its image, etc. It points out the importance of European cultural influences on the urban image and culture of the city. After that, one thinks about self-awareness, which determines whether the tradition will be preserved and passed on to the next generations. The level of cultural competence depends on self-awareness [1]. The high level of cultural competence [2] is part of the image of Novi Sad. Respondents assess the impact of the sudden and large influx of new population over the last decades [3] on the city's cultural competence.

Novi Sad won the prestigious European title 'European Capital of Culture 2022'. In this regard, the paper seeks to show the reactions of the population of Novi Sad. Respondents commented on their impressions of the image and cultural changes in urban areas caused by the COVID-19 pandemic. The significance of the work is that special attention was paid to the positive outcomes caused by the COVID-19 pandemic. At the end we discuss critics

of the urban image and the cultural occurrence and phenomena. The main hypothesis explored in the paper is that negative phenomena, such as a pandemic, can result in positive effects on the urban image and visible cultural changes.

## 2. Theoretical Background

An image of a place is the first association with it when it is mentioned. However, in the scientific literature, it is mentioned more sporadically. It is very important to note that the authors understand the term image as visible, but also are aware of those intangible symbols. In addition, it is important to note the factors that have had or still have an impact on the characteristics of the image of the city.An urban image consists of possible forms of urban buildings [4,5], styles, forms [6], and architectural solutions, but respondents mentioned this concept before thinking about how they perceive the overall image of the city. Changes in the urban image are mentioned in settlements where major spatial transformations occur, such as in the work El Amrousi & Elhakeem [7]. A sustainable urban image links the physical character of the built environment with the environmental, social, economic, and cultural aspects of that environment [8]. Accordingly, Fuli [9] writes that urban image includes the material and the intangible heritage such as inter alia, habits and spirits of citizens. Symbols of urban history preserve the image of the urban landscape in the frame of identity and community cultural values [10]. Social change and economic transformation reshapes the city's spaces and image. Cultural initiatives can cause heritage-creativity hybridization in the city [11].

Non-physical factors affecting any urban image must be linked to visual imagery to present a coherent city image [8]. Regional culture as an important core content reflects the characteristic value of the city [12]. The muteness of the region of Vojvodina [13], whose capital is Novi Sad, was, among other things, colorfully reflected in the rural area. The settlements with a majority Slovak population were dominated by blue [14], Hungarian by green, Croats by red [15], Romanian by golden yellow [16], and so on. This historically conditioned heritage explains why Novi Sad, where all ethnic groups of Vojvodina are present, is most colorful (Figure 1). Florenzano et al. [17] confirm chromatic interventions in cultural heritage have not been overcome and should be discussed in its historical, theoretical, and phenomenological complexity.

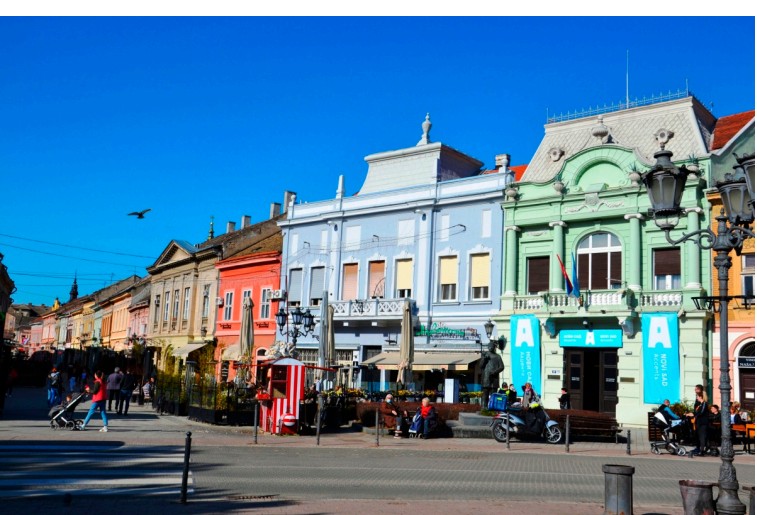

**Figure 1.** Colors of ethnic groups on the facades of one of the oldest streets in the city. Source: Bojan Đerčan, February 2022.

Presenting, recognizing and recommending the city's heritage creates new employment opportunities for the local population [18]. This work can influence the correction of urban regulations which could improve the image of the city.

### 3. Methods

Available literature was used for historical facts. Basic information about the European Youth Capital 2019 and European Capital of Culture in 2022 has been downloaded from official documents and from the Internet. The research part of the work was carried out using qualitative analysis and by applying deep interviews. Only with a qualitative approach to data collection can the attitudes and answers of the respondents be investigated in detail [19].

The deep interviews were conducted at the end of 2021. Because of the restrictions imposed by COVID-19, some of interviews were conducted by phone, email, social networks, as well as in a physical format. The duration of the interview with respondents was on average 15 min. As in a research paper of Kovačević et al. [20], during the process, the interviewer made his notes and observations to facilitate the coding, analysis and interpretation of the data. All participants were residents of Novi Sad.

Respondents contributed to the objectivity of the research by giving their opinions on the set topics. According to Sapu [21] and Cvetković et al. [22], interviews and focus groups are highly beneficial for gathering detailed information about people's values, beliefs, and opinions, and for finding out how a 'group' or perhaps a community feels about a particular issue. Respondents had to answer five questions related to socio-demographic characteristics and five questions based on the theme of the paper. The respondents' conversations were recorded on a mobile phone, and the most useful and expedient answers were selected during the desk research. The most constructive answers are given in the paper.

#### 3.1. Questions

The first question was intended to find out what the residents of Novi Sad think about the influences on city image and culture in the broadest sense of the word. Did they see the historical background? Do they think of the recent or distant past? Do they feel the effects of globalization? Do they recognize the uniqueness that arose on the spot and that is not conditioned by anything?

The second question examined self-awareness, which is very important. Without it, there can be no originality. Self-awareness adds an intangible to the image of the city. Culture has no future, because the influence of tradition is lost. More precisely, there are always traditions, but the question arises: whose is it and what kind of tradition is it?

The third question explored self-awareness about the current situation. How aware are the respondents of the title of European Capital of Culture? What does the new title mean to them? Does it affect the image of the city?

As everything is happening in the midst of the COVID-19 pandemic, one of the questions is dedicated to noticing its impact on all cultural segments. What has the COVID-19 pandemic changed about the city's image, if anything?

The fifth question deals with the identification of the shortcomings with regard to the city's image. Respondents also offered possible solutions to some of them. Pointing to others should in this way mobilize intellectual forces in an attempt to overcome them or at least reduce their intensity.

#### 3.2. Sample

The researchers sought participants who fit the following three criteria: 19 years of age or older, residents of Novi Sad, and have at least a primary education. The selection of respondents for the in-depth interview was carefully undertaken. Two respondents are people who are active in different ways in the culture of the city of Novi Sad. Others were chosen to be of different ages, backgrounds and educations. Participation in the study was voluntary. The sample consisted of 10 women and five men. The absolute majority were born in Novi Sad and they are highly educated (Table 1).

**Table 1.** Socio-economic structure of respondents.

| Socio-Economic Structure | Number | Percent |
|---|---|---|
| Gender | | |
| Male | 5 | 33.3 |
| Female | 10 | 66.7 |
| Age | | |
| 20–29 | 3 | 20.0 |
| 30–39 | 3 | 20.0 |
| 40–49 | 3 | 20.0 |
| 50–59 | 3 | 20.0 |
| 60+ | 3 | 20.0 |
| Origin | | |
| Born in Novi Sad | 8 | 53.3 |
| Inhabited | 7 | 46.7 |
| Education | | |
| Primary school | 1 | 6.7 |
| High school | 6 | 40.0 |
| Faculty | 8 | 53.3 |
| Total | 15 | 100 |

Source: Authors' findings.

In abbreviation, next to the answers, there are signs that indicate the basic socio-demographic characteristics from the respondents. R means respondent, followed by his ordinal number, than gender (m—male, f—female), and age. The penultimate word refers to the origin (b—born in Novi Sad, i—inhabited). The last letter indicates the educational level of the respondents (p—primary school, h—high school and f—faculty).

## 4. Results and Discussion

The deep interview consisted of five questions. The most interesting results are presented in the paper and they indicate differences in the perceptions of the population of different socio-demographic characteristics.

### 4.1. What Can You Say about the Influences on City Image and Its Culture?

"Novi Sad has formed its unique identity under the influence of Europe. Since the formation of the city, various European nations have lived in it and with its culture and tradition has influenced of its multicultural originality and unrepeatability. (R1f45bf) It is no coincidence that the most pioneering and ground-breaking advances in human development originate from cultures that embrace diversity as a positive challenge rather than as a threat [23]. Multiculturalism has formed an urban image of Novi Sad.

"Europe influenced the architecture of the city (the old city core), the formation of cultural institutions, the mentality of its first inhabitants and their descendants, etc." (R2f70bf) Every occurrence in the present is historically conditioned [24]. Matica Srpska (1864) was moved from Pest, (which is part of Budapest and where were established in 1826) to Novi Sad. From 'Matica Srpska' emerged the most important literary and cultural institutions of the Serbs, like the Museum of Vojvodina (1847), the Gallery of Matica Srpska (1847 in Pest) and book trade and publishing companies [25].

Respondents recognized different influences: east, west, plains, hills, north, south... "Something is constantly being copied from the west. There are good things there, such as music. The bad thing is that movies raise our children". (R8m54ip) "In Novi Sad there are also influences of the east, such as in the architecture of Orthodox sacral objects, houses of individual housing, gastronomy, etc." (R5m48bf) 'The population of Novi Sad was previously recognized according to manners that it accepted from Europe and which distinguished it from the population from the mountainous regions of the Balkan Peninsula.' (R4f59bf) The age of the respondents of this group of answers is quite close in relation to others. This fact indicates similar educational forms that existed at the time of their education.

"Oh, that's hard to say. Each new time brought some of its own people. Some of them merged with the environment, and some, by bringing their culture, enriched the city image." (R3m63bh) According to Lawler [23], differences make the cultures of the world richer and it is not in anyone's interest to let anybody give up their culture or their identity.

The answers of the following respondents also support this. "Traditionally, Novi Sad closely follows cultural events in Europe. Classics are listened to and played in Novi Sad; but also the accordion. It reads William Shakespeare, Honoré de Balzac, Chekhov, but also Orhan Pamuk. So, in Novi Sad, he strives for the image of a city that respects quality and progress, no matter which side he comes from." (R7m38if) "The culture of the city is seen at every step and it is diverse. There is everything: ballet and boxing, courtesy and vulgarity, and cleanliness and dirt." (R15f33if) These two respondents come from the same age and educational category. They are in their thirties and have university degree. Also, they were not born in Novi Sad.

"The Internet is the cheapest source of information and all cultural influences on young people come with it. The Internet should be used to improve the city's image" (R12f21bh) Global culture comes through different media and it doesn't matter where you live. Intellectuals are culturally sophisticated, and their perception of cultural influences is related to their interests. (R13f28if) The youngest respondents mentioned globalization. From this it can be concluded that it was recognized during the 21st century. According Ritzer & Dean [26], globalization has been recognized in various moments of human history and various segments of life. According to the opinion of Pieterse [27], the last wave of globalization arrived at the end of the 20th century. This author mentions three paradigms: growing homogenization, cultural convergence, and the global mixing of cultures that has resulted in a global hybrid culture. It was also recognized by the respondents.

Some respondents gave unpleasant answers, which resulted from their bad experiences (historical, financial or those of some other nature). "It should not forget the days of the European culture that we suffered in 1941, 1942, 1944 and 1999. Then Europe distorted the image of the city". (R6m42ih) "Young people see only a 'culture of survival'. Low income, high costs, expensive food. I go to the theater, wait for the show to start, and then I beg to be released without a ticket. So, whoever fights for the crumb of culture cannot define who has the decisive influence on the city image". (R14f25ih) "Many cannot get rid of the culture they came from, so they are recognizable by their noise, disorder, negligence and arrogance. Unfortunately, that also fits into the culture of the city and tarnishes its image." (R9f51bh). "Unfortunately, you are researching what affects the image and culture of the city, and not how it changes immigrants. I conclude that you assume that decades of suffering are her tameness and non-aggression." (R11f65bf) Theespondents do not share any common socio-demographic categories. This means that each of them recognizes something bad that they must mention and that these things are different from the historical moment in which they were recognized.

"I liked this city and that's why I stayed in it. Calmness, courtesy and mutual respect. But, I also remain to change, to progress, to accept everything we like, regardless of where it comes from. I think it improves the quality of life." (R10f38ih).

The European influence is in Novi Sad is indisputable. The inhabitants of Novi Sad are constantly looking towards Europe and they are happy to accept new cultural achievements. Politics in the Second World War and in the spring of 1999 adversely affected the experience with some European countries. A fraction of the population who lost family members feels a natural aversion to the causes of their family disasters. Destroyed bridges and buildings have changed part of the physical image of the city. The rebuilding corrected the appearance of the city's symbols.

An example of representative coexistence in a multiethnic environment is one of the most recognizable images of Novi Sad. Similar multiethnic environments in the world, those are not at that peacetime level, could look up to that image. In the questions on influences on image and culture, the thematic grouping of answers according to the age of the

respondents was recognized, but also according to their origin. The material symbols of the city's image are definitely the safest guardians of its history, tradition and cultural heritage.

*4.2. What Do You Think about the Self-Awareness of History, Tradition and Cultural Heritage?*

The image of the city is formed by history, tradition and cultural heritage. That is why the self-awareness of the residents regarding these mentioned things is very important. A focus on self-awareness is necessary for understanding and developing responsible personal relationships with those who are similar and different from oneself [28]. Strengthening people's cultural awareness and competencies in ways that are represented by respect for both one's own and other cultures according to Yang & Gao [29] is extremely important in multicultural environments such as Novi Sad. Depending on how much someone is aware of their history, tradition and cultural heritage, it depends on how much they will pass it on to younger generations, that is, to nurture and preserve it.

"We were listening about famous institutions, buildings, people, and customs from the teacher. Novi Sad is growing, building, changing and all that is now history. I gladly remember when I hear someone talking about it on local TV." (R12f21bh) "The cultural foundations on which the identity of the city has been built can not be found in the curriculum of elementary schools. How can we expect that the young generation will have some self-awareness about tradition, culture, and cultural heritage?" (R1f45bf) "Being aware of history, tradition and cultural heritage means to have ancestors who lived in Novi Sad in the past." (R2f70bf) "Self-awareness comes from the family. I heard about old Novi Sad from the lady I lived with while I was studying. After that, it is nowhere to be found." (R13f28if) What these respondents have in common is that they are women. Thus, they are broader and deeper in terms of self-awareness in relation to male respondents. Women should be in politics, as Lukić et al. [30] mentioned, because they will work towards the self-awareness of young generations.

"Vojvodina, even Novi Sad, as its largest city and administrative center, has always been called 'Europe in small.' I am very proud of that". (R3m63bh) This is because different ethnic groups inhabit the Province of Vojvodina [31]. The results from Šagovnović [32] also recognize the presence of a sense of pride of "being from Novi Sad".

"The natives of Novi Sad are aware, but the city has been inhabited by many immigrants, so the purpose of history, tradition and culture has slowly become meaningless." (R4f59bf) Bubalo Živković et al. [33] talk in detail about the settlements of this area. The presence of a new population brings new details to the urban image of the city.

He et al. [34] have noticed that the cultural heritage administration and the residents play a role in preserving the culture of the historical sites, which promote sustainable preservation. "Since the middle of the 20th century, industrialization and socialist development, then at the end of the same century, political and economic changes in the former SFRY (Socialist Federal Republic of Yugoslavia) have influenced the immigration of the population that is not familiar with the past of the city. They do not feel the spirit of Novi Sad." (R5m48bf) Similar examples are evident in Europe. Beeksma and Cesari [35] write about a museum staff member, between whom they found social cleavage between those who feel that they belong and see themselves as locals with roots and those who don't.

"We are aware of famous people who are the basis of the tradition and culture of Novi Sad. Starting with George Balasevic (musician), Miroslav Antic (famous poet), to Mileva Maric Einstein (scientist and wife of Albert Einstein) and Jova Zmaj (children's poet, but also a doctor)". (R9f51bh) Scientists are already leaving traces of these greats. Their names are mentioned in the works of Tadić and Đurđević [36], Bojkov [37], Esterson and Cassidy [38], and Stošić [39]. They are generally known in Serbia and the former Yugoslav republics, so they are a symbol of the intangible image of the city.

"I was not born here, but the culture of living is close to my hometown. Everyone loves dogs, everyone can ride a bike, everyone is on the banks of the Danube during the summer, everyone is procrastinating when talking and not rushing when eating. It's all Novi Sad." (R10f38ih) This is one of the images of the city that describes the dynamics of

life, the spirit of the inhabitants and the 'atmosphere'. "Yes, yes I am a hedonist and I am aware of the traditional culture of eating. Novi Sad is a legendary place with good wine and food." (R6m42ih) According to Figueiredo [40], food is cultural heritage and it can be the way of promoting some territory. Respondents who tied their self-awareness to life culture styles are of similar age and education (they have a high school diploma), but were not born in Novi Sad.

"I heard about Matica srpska, great patrons whose money was used to build palaces, churches, high schools. However, music is my life and for me the heritage and tradition is EXIT, the Festival of Street Musicians, or 'NOMUS' (Novi Sad Music Festivities)". (R7m38if) Sacral objects of different confessions provide adequate offer for tourists of different religions [41]. The EXIT festival is the most important promoter of the Novi Sad. It is held annually during the summer season, and attracts a large number of young tourists [42]. In January 2018, EXIT was once again crowned Best Major Festival at the 9th annual European Festival Awards, which took place at Groningen's De Oosterport in The Netherlands. EXIT won the Grand Prix in a fierce competition comprising of over 350 festivals in 35 countries [43]. It proves that Novi Sad has become the center for youth all around world. The EXIT Foundation initiated and led the process of the candidacy of Novi Sad for the European Youth Capital 2018. The youth organizations of Novi Sad continued the process and brought victory to Novi Sad [44]. Festivals such as EXIT have an impact on increased tourist traffic. Tourists contributed to a new phase in the formation of the city's image, but they also promoted it upon returning home. Therefore, events are an instrument for modifying the image of the city. Other musical events can be found in the works Novaković and Mandarić [45]; Doğan and Simolin [46] and others.

"I think my generation is very aware. We were raised in a spirit of tolerance, intercultural respect and esteem. It all starts with the Petrovaradin Fortress, tamburitza orchestras, quay, beach, and ends at the Gallery Square (if we think of art), Spence (sports) or at the Serbian National Theater, Benakiba or Youth Theater (theater culture)". (R11f65bf) That these are the greatest potentials of the city is confirmed by the works of Reba & Kostreš [47]; Konstatinović & Jović [48] and Kopić & Polić [49]. The Petrovaradin Fortress is a multilayered archaeological site. In addition, it is a master of the work of the famous architect Sebastien Le Prestre de Vauban, a French military engineer. The fortress is connected to numerous historical stories, but also legends [50]. Oral and written lectures have not yet been put into the form of tourism trends that would have a financial benefit. This group of respondents recognized the history, tradition and cultural heritage in the sights that are material symbols of the city's image.

"Yes, I respect. I notice that living legends, symbols of Novi Sad, walk the city (actress Mira Banjac, poet Pero Zubac). They haven't even the part of respect they will receive when they are no longer alive, as was the case with musician Đorđe Balasevic". (R15f33if) Ekelund et al. [51] confirm this claim. They argue that the works of art of those artists who are no longer alive are more expensive.

"I have moved in here recently and I do not feel obliged to know anything special now. I know something, such as that there are a lot of people of different origin and that everyone lives in peace." (R14f25ih) "Do not ask me. The past has passed. For me, the past is painful and has nothing to do with Novi Sad. I am only interested in the future." (R8m54ip) Research has shown that there are those who were not born in Novi Sad and who do not feel the need to have any level of self-awareness about its culture, tradition, and heritage. Such answers were not expected. Due to their strangeness, they had to be involved.

Some respondents have self-awareness, but obviously not everyone has acquired and recognized it. Based on this answer, it can be concluded that the past of the respondents affects the recognition of the image and interest in the cultural values of Novi Sad. Thanks to its cultural heritage, which has been created for centuries by European influences, Novi Sad can be included on numerous European routes, such as the gastronomic, Danubian, and Voban heritage routes, tolerance routes, festivals, etc.. They can be in the function

of preservation and economic sustainability from the city, without which ecological and population sustainability cannot survive.

*4.3. How Does the Title 'European Capital of Culture' Affect the City Image and Cultural Situation?*

The "European Capital of Culture" has been chosen for 35 years and it is one of the most important cultural initiatives in Europe. Cities are selected on the basis of a cultural program that must have a strong European dimension, promote participation and active involvement of the community, but also support the long-term development of the city and the region. Novi Sad is the first city to take the title of "European Capital of Culture" within a special program for candidate countries for membership in the European Union. The COVID-19 pandemic postponed the title from 2021 to 2022, as in other cities in Europe with the same title [52,53].

The European Capital of Culture 2022 is a platform for the development of the creative potential of Novi Sad [54]. This project should motivate and inspire both cultural workers and all citizens to re-examine current values and set new goals toward the democratic cultural development of the city. The re-examination of the modern identity of Novi Sad, the revitalization of its cultural heritage, the reconstruction of the existing and opening of new spaces intended for culture, and the developing of cultural participation of citizens are just some of the principles of cultural development [55].

The concept of Novi Sad's bid—'For New Bridges' used the bridge metaphor as a connection, building upon the symbolic meaning of the city's bridges over the Danube which were built, destroyed by wars and then reconstructed. A recent trauma in the city's memory was represented by the 1999 NATO bombings, when all three bridges were destroyed. Today, they represent strong "lieux de mémoire", reminding people of the strong sense of solidarity of the local population who, back in 1999, tried to protect the bridges against the bombings at the cost of their own lives. The concept also has symbolic connotations in the context of Serbia's European aspirations for joining the EU, hence, new bridges needto be built. Belonging to a non-EU country, the title represents a way to 'reintegrate' the city and Serbia "into Europe's cultural life, through a dialogue of cultures" [56]. By promoting the city as a cultural destination, a young city of culture, peace and reconciliation, it was aimed to stimulate citizens' pride [57].

The "New Town" is an urbanistic and architectural competition for the financing of small public spaces. Numerous arrangements and reconstructions of public spaces, so-called new places for meeting citizens and cultural events, are part of preparations for 2022, when Novi Sad will be the European capital of culture.

'Microgranting—small town' is a real competition in order to raise awareness about the significance of the participation of the local community in the organization of small areas of public interest. Another goal is animation of citizens and the local community to organize small urban spaces through joint work.

The "Audience in Focus" program, which represents the first phase of the realization of the project platform from the Application Book "Outside the comfort zone, aims to strengthen and support cultural institutions in Novi Sad and their continuous work on the development of cultural habits, needs and competencies of the citizens of Novi Sad. It is a financial support for the realization of projects of citizens" participation in public cultural life. The answers below show what the residents of Novi Sad think, and how the title of "European Capital of Culture" affects the city image and the cultural situation

More positive responses were obtained from people who deal with any kind of culture. This reveals that the essence is in information and interest for the culture. These respondents are women of similar age and who were born in Novi Sad. "I think it's good that this happened to the city. It is an opportunity to draw attention to multiculturalism, more decennial interethnic tolerance, the benefits of cultural diversity, strengthening the European spirit of the city. This specialty can be put into the function of tourism, which can manifest with different bonuses in the future." (R1f45bf) The title is a huge positive

impulse for improving the quality of the city's image. "Since its inception, Novi Sad has been a cultural Mecca. It deserved to show his qualities to the whole world. I think this title is an opportunity for that." (R9f51bh)

Some answers, given by those born in the city, show that the title is important for young people. "I watched the writing process for a competition that brought the city a prestigious title. I have to point out the great efforts of the team that worked on this job. Creative young people should invest their potentials in exploiting old (perhaps forgotten) and creating new unique cultural results." (R2f70bf) "If this will bring young people into the city, then it is great at this time of 'white plague' (namely at the time of insufficient birth that reduces the potential size of the population) (R3m63bh) "Super. We young people love different events. If they are free, the better. We are happy. We look forward to the beginning. We will attend every event." (R12f21bh) "Bravo, for those who managed to bring it to our city. They will probably know what to do with it. I would use that to promote cultural institutions. I would cancel the tickets. Entrances would be free, as in London." (R4f59bf)

Observing the benefits of the title of "European Capital of Culture" from the financial aspect came from respondents who are in the third or fourth decade of life. "Only positive, just as befits the city. Public buildings in the city center are being cleaned, repainted and tidied up. Manifestations, performances, concerts, guests are announced." (R5m48bf) "I see that the facades in the center of the city are being renovated, the squares are being arranged and more work is being done to promote the actions of the city government." (R15f33if) "If this city title brings some money, I hope that it will be used in the best possible way." (R10f38ih) "It is very important for the image of the city, tourism, the present, but also the future. The goal is that almost every citizen giving the contribution, it can feel back the material or spiritual grace". (R7m38if)

"The title affect on the city image in various ways and useful. For example, the British magazine *Time Out* put Novi Sad on the list of world capitals that offer unusual cultural events." (R13f28if) *Time Out* [58] writes: "Novi Sad has gone all in ahead of more than 1500 events featuring 4000 artists, including an exhibition in The Mlin Cultural Station, an abandoned pasta factory. Many have been making the pilgrimage to EXIT Festival for years, but 2022 will see Novi Sad's gorgeous architecture and unique history put it on the map as a major destination-in-waiting".

Female respondents expressed concern for the cleanliness of the city. "I expect that this will increase the city's cleanliness." (R11f65bf) A skeptic said: "Novi Sad as 'European Capital of Culture' will increase number of events. It looks like as the EXIT festival. Some will be enriched, and most will be the witness of dirt that needs to be removed." (R14f25ih)

Some answers were not clear, nor precision. "I heard somewhere about that." "I saw some advertising at the wall of the shopping mall and in the centre of city. There is nothing wrong with that." (R6m42ih) "I heard something. Great." (R8m54ip) Such answers came from male respondents.

Most respondents recognize and respect the title. Kovačić et al. [59] did not find any negative perception or influence of the title "European Capital of Culture" with a detailed analysis of the literature. Respondents of this exploitation find it easier to notice changes in the streets (in the inner physiognomy of the city) than the implications that will appear in some later cultural phenomena. People like the good news and look forward to it. They believe that it is already having a positive effect on the city's image and situation in it. Some respondents have heard of the title, but are unsure how it could affect the city. Obviously, there is a need for more intensive marketing activity. Flyers should be provided and distributed freely, for example with electricity bills. They should point out everything that the title brings, from which every citizen of the city could find some benefit. The suspicion of the respondents is based on lack of information or on bad experiences. It can be a good incentive to work on the transparency of the performance of persons in the project team who won the title.

*4.4. In Which Segments of the City's Image and Its Culture Do You Recognize the Impact of the COVID-19 Pandemic?*

The pandemic had the greatest impact on segments of everyday life culture. The culture of life is changing spontaneously. The COVID-19 pandemic has spread and raised the level of self-awareness about the need to maintain hygiene. It tacitly set the standards. The imposition of mandatory wearing of masks in public places has changed the general impression.

Respondents emphasized wearing masks in the first place. "The mask has become a cultural standard 'overnight'. Masks are mandatory in public places. The hosts pay fines if a visitor without a mask is spotted. However, this rule is not followed whenever feasible. So, there is a lack of self-awareness or conscience." (R1f45bf) "Under the mask, I feel unpleasant smells around the city better than before." (R9f51bh) "If the image of the city is the image of its inhabitants, then I see changes in its face. Masks, visors, gloves, disinfectants . . . Since the beginning of the pandemic, I feel like I'm constantly at a masquerade ball." (R6m42ih) The masks were mostly discussed by female respondents over the age of 40. Milošević et al. [60] wrote that the local governments of Novi Sad should promote urban park and river quay usage during pandemics with a major focus on protective measures such as physical distancing or limiting of the number of visitors, if necessary, rather than having mask mandates that cause discomfort to the population. Dragic et al. [61] confirmed that lockdown in Novi Sad caused the positive effect of improved air quality on public health.

"The culture of living has changed for the better. People began to pay attention to hygiene. In order to avoid crowds in public transport, bicycles began to be used en masse. I am a big fan of Amsterdam. I am glad that Novi Sad got a cycling image." (R11f65bf) People began to avoid all places where they could be found among many people. There were primarily means of public transport, i.e., buses. Although there have been bicycle rental points since 2010, it seems that with the COVID-19 pandemic, their use is becoming more widespread. Relief (alluvial flat land and river terrace) and climatic characteristics (moderate continental climate) of Novi Sad are ideal for this activity. From 2021, the city subsidizes the purchase of bicycles, renews bicycle paths, traces new ones and thus emphasizes alternative ways of moving. In addition, it is a method that fights traffic density, solving parking spaces and the quality and health of the population. The work on popularizing the use of bicycles and educating about its positive aspects, in the time of the COVID-19 pandemic, gained special significance. It has been proven that the spread of the infection is less in open spaces. Komadina [62] writes that people who cycle expressed their satisfaction with the number of parking spots, storage space at home, safety in traffic, and the quality and density of cycling paths. Cycling will definitely be a puzzle in the future in the complex image of the city.

The perception of the increased number of cyclists on the streets was contributed by the appearance of food delivery people and other similar courier services [63]. In this way, people helped the hospitality and restaurant sector to overcome the losses caused by the recommendations on avoiding public places and social contacts. On the other hand, they are welcome to the elderly population who have difficulty moving and are afraid of infection. Also, a significant number of unemployed people thus became useful to society and found a source of income. One of the interviewees emphasized that "the COVID-19 pandemic has developed a business of delivering food and other necessities. For me, it really simplified and made life easier." (R3m63bh).

Another respondent saw a positive pandemic phenomenon. "I recognize the impact of the COVID-19 pandemic in many segments of the city's culture. I felt that there was mutual understanding, respect and the need for solidarity. You will agree that these are cultural achievements that have been neglected." (R13f28if) "In the culture of communication, the unavoidable topics are vaccination and how did you survive COVID-19?" (R14f25ih) Sympathy with the problems of the elderly, but also their peers, was expressed by the youngest respondents. It is clear that the COVID-19 pandemic will disrupt daily life until

the majority of the world's population is vaccinated or until effective medical treatment is available [64].

Most of the answers indicated how the COVID-19 pandemic indirectly took people out into the fresh air. Culture, art, and festivals can all be used to boost a territory's attractiveness and inventiveness [65]. "COVID-19 disrupted the culture of living, established celebration ceremonies, thinned out visions, and alienated people." (R12f21bh) "The COVID-19 pandemic forced parents to devise birthday celebrations in nature, animating children for outdoor games." (R10f38ih) "The COVID-19 pandemic has displaced numerous cultural events outside the confined spaces. Then they became more visible to larger groups of people compared to the time before." (R15f33if) From 2020, Chinese lanterns are placed every winter in the 'Liman' Park [66]. They symbolize the friendship of the two peoples, but also the presence of the Chinese in the city. Compared to other smaller ethnic groups, the Chinese are the shortest present on the territory of the region of Vojvodina, but have a transparent influence. In addition, manifestations similar to this one (Figure 2) fill the gap in cultural life of the city created during the state of emergency and curfew that were introduced in order to prevent contacts and reduce the spread of COVID-19. The appearance of culture on the streets in the form of manifestations, exhibitions, performances certainly beautifies all aspects of the city's image.

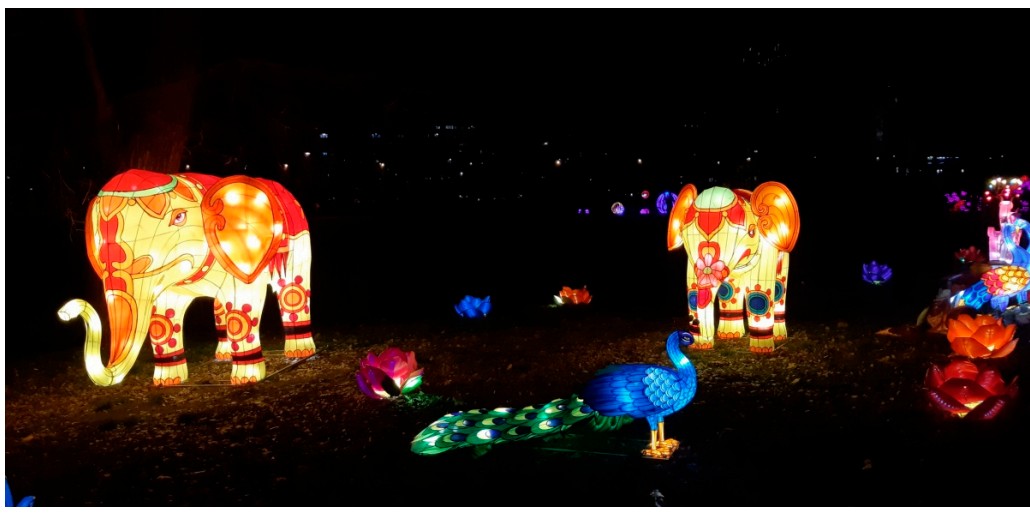

**Figure 2.** Chinese lanterns in the 'Liman' Park. Source: Bojan Đerčan, Ferburary 2022.

Culture that is impossible to organize on the street has been moved to the virtual world. "Culture began to be followed over the internet. Some institutions, such as the 'Matica Srpska Gallery', provided virtual visits. Some have made short films, such as 'The Museum of the City of Novi Sad' about the part of the Underground military galleries that are under their protection. They are available on YouTube". (R5m48bf) Although the literature records the digitization of cultural sites even before the pandemic [67,68], it seems that they gained complete purpose during it. "The pandemic has negatively affected my culture. I worked from home. The children went to school 'online'. I had problems with the internet connection. In the house, we all fought for the computer. I started swearing often. I want it never to happen again." (R7m38if) "The culture of living has been digitized. I drink coffee with a friend via Skype, I meet via Zoom, I read e-books, I pay bills via e-banking, I buy food via social networks... Then I wait for the health consequences of sitting in front of a computer." (R4f59bf) The functioning of civilization at a distance was mostly discussed by respondents who have a university degree. Turning to the virtual world has imposed the need to present the image of the city in that area as faithfully as possible. For that, the most valuable help is those who are the most educated.

Some respondents did not see anything good in the pandemic. "Closing in houses and restricting the movement of people over the age of 65 seemed like great discrimination.

Music culture saved me from pandemics." (R2f70bf) Cancura [69] approved that music offers perspective, it heals, and it allows for compassion—all so that each and every one of us can lead a big and full life. "The culture of living was lost because of survival. The cult of death, fear and lack of money were brought by the pandemic. Everyone ran away and there is no company to drink beer. How, then, can I culturally and politely ask someone to lend me money?" (R8m54ip) The work of Kalenjuk Pivarski et al. [70] indicates that the ban and restriction of movement in certain phases of the COVID-19 pandemic has led to a reduction in visits to catering facilities. For someone, it changed cultural habits, and for some, a complete lifestyle.

According to the respondents, the COVID-19 pandemic brought masks. The pandemic imposed the need for distancing, which was facilitated with bicycles. On the one hand, it moved culture and art from the closed to the open space. On the other hand, the pandemic closed the culture in a virtual space, opening up endless possibilities for it. The andemic influenced the recognition of the need to help the helpless and provided an opportunity for more mass organization of such services. The pandemic provoked interpersonal understanding and solidarity, but also fear for one's own life.

*4.5. List the Biggest Deficiencies of Image and Culture of Novi Sad?*

"The number of inhabitants in Novi Sad is growing, but the number of cultural institutions does not increase." (R1f45bf) During 2021, two cultural institutions, the Ballet and Music School, will receive a new building with the largest concert hall in the region. World-famous violinist Stefan Milenković is returning to Novi Sad to give his contribution to the cultural advancement of the city [71]. This is a good criticism, because a city with 231,798 inhabitants 2011 [3] needs many more different cultural institutions.

"Information boards should be placed in front of every landmark of the city of Novi Sad." (R7m38if) The boards would have an informative function for tourists, but also an educational function for new generations. "I would form cultural points. They would have an advisory function. They could, for example, advise teenagers on how to trace their energy, ideas and free time in various forms of cultural enrichment." (R5m48bf) The most educated male respondents suggest ways to more aggressively share information and information for educational purposes of different groups.

"There is a lack of water parks, theme parks, classic parks to enrich the culture of living, especially of young people." (R12f21bh) From 2016 to 2020, Novi Sad had the first theme park, Dino Park [72], modeled on similars in Europe (Saint-Hilaire-de-Riez, Agde, Charbonnières-les-Sapins). However, a tumultuous past can be inspired by many similar, such as Puy du Fou in France [73], historical theme park with epic, period-specific shows. According to Conić [74], the city really lacks a water park and its construction is planned. They will complete and beautify the city's image in the future.

"Novi Sad needs to enrich the culture of living with green roofs and fountains." (R13f28if) Green roofs and green walls are effective resilient and adaptive solutions that provide multiple ecosystem services when implemented in urban environments [75]. The paper of Jeftić et al. [76] explains how much green roofs Novi Sad needs and that it is necessary to start their construction as soon as possible. Novi Sad, with an urban area of 129.7 square kilometers [77], has only four fountains and many drinking fountains. They positively affect the microclimate conditions, and attract users to their surroundings [78]. Drinking fountains are very important significance and associations which are related to the culture and meetings of people in region of Vojvodina [79].

"An original attraction should be made, such as an elephant in Nantes or a spider in Liverpool." (R15f33if) This remark is constructive and may be the subject of some further research on what could be the symbol of the city.

More comments indicate noise. "Communal police cannot silence loud music. I feel a lack of culture of living." (R3m63bh) "Our people are inclined to accept something that is not ours. For example, the Chinese believe that noise drives away evil. In our country, children like to make noise with firecrackers. They usually do not know what they are for.

They are not our cultural heritage. This is not taught in school. Firecrackers can hurt them. They make noise." (R4f59bf)

"I'm bothered by the lack of culture in traffic. Cyclists are carefree and self-sufficient. People on scooters are arrogant. Taxi drivers are rude. If the street has more lanes, the one closest to the sidewalk is occupied. There are too many swear words, sirens and noise." (R6m42ih) Đerčan et al. [80] wrote about traffic noise. The new bridge [81] is expected to reduce congestion and noise in the city.

"Newcomers live where there is garbage under the windows. The natives are negligent towards existing buildings, squares, green areas, and often even monuments." (R2f70bf) The literature proves the testimony of this respondent. Krklješ et al. [82] noticed the negligence of institutions and organizations that should intervene in public spaces. Dragin et al. [83] talk about inattention to the cleanliness of the Danube bank. "Novi Sad lacks garbage cans, especially in residential areas of the city." (R8m54ip) Dragić [84], Aleksić [85] and similar wrote about this problem. "Some show their non-culture when they leave dog feces in the streets." (R9f51bh) Children fight against this phenomenon by tying bags to trees. The bags are intended for dog owners to collect their feces. Children say not to soil the surface for walking or playing. The younger generations offer hope of preserving environmental awareness. "There is little space for dogs to run" (R10f38ih) According to Stakić [86], there were 19 locations for running dogs in Novi Sad. Seven are decorated according to the highest European standards. All respondents were extremely critical of the quality of culture of living on city streets.

"Young people destroy urban facades, so urgent need to find out mechanisms for eradicating this phenomenon. It seems to me that this can damage the city's image." (R11f65bf) This phenomenon is known in the Balkans [87]. The daily press testifies to vandalism that can be in the colors of the flag [88]. However, the memory of the famous inhabitants of Novi Sad brings murals dedicated to them [89]. It is like a noble compromise that channels creativity and the need of the spray master to satisfy the need for his cultural expressions. According to Cercleux [90], graffiti is considered to be the starting step of street art development, therefore referring to an art that can bring a value to the city and is made on the basis of authorizations or the acceptance of the owners of the concerned places.

"Political influences should be excused from the cultural life of the city. The existence of only cultural policy is desirable." (R14f25ih) Volić [91] writes in detail about cultural policy. The politicization of culture [92] is a frequent topic in the daily press, and it is even mentioned in the Strategy of Cultural Development of the City of Novi Sad [93].

Respondents are aware of the biggest deficiencies of culture in their city (lack of cultural institutions, information boards, different type of parks, fountains, original attraction) communal problems (negligence, vandalism, noise, garbage, and communal waste), politics influences and cultural policy.

Great self-criticism left out positive examples in some of the respondents' answers. The streets of Novi Sad are clean, so everyone is terribly upset when it is disturbed. In addition, the privatization of housing has led to increased concerns about the appearance of facades, entrances and driveways. This is supported by the prescribed communal penalties for clearing snow, disintegrating the facade or similar transgressions.

Respondents think that deficiencies were created among the population of a certain age. They have some constructive proposals for their overcoming and eradication. "Act locally—think globally" [94] is a solution to the problem, if the themes related to the culture of living in the city would be included in school plans and programs at the most serious level. The culture of living is not of a political nature, it has nothing to do with conflicts and it certainly does not harm [95] and therefore can be unreservedly propagated. The positive thing is in the responses is that everyone sees gaps in everyday life (which can be repaired), not in cultural institutions or in some higher forms of appearance, such as art (in which it is not expected). It is the most important to have self-awareness about how to preserve the image of the city, as well as ideas for its improvement.

The culture of living is not of a political nature, it has nothing to do with conflicts and certainly does not harm [95]. The positive thing in the answers is that everyone sees gaps in everyday life (which can be repaired), and not in cultural institutions or in some higher forms of appearance, such as art

## 5. Conclusions

Judging by the answers of the respondents, Novi Sad built its image more on intangible than on material values. These are primarily relations of multiculturalism. Respondents recognize historic buildings that are the image of the city. They feel the global influences coming through the Internet. They also believe that we should react in the opposite direction. Using Internet, it should to place a positive image in the same way.

Women have distinguished themselves as people who have a greater self-awareness of history, tradition and cultural heritage. How to persuade people to respect the city they moved into and to inform themselves about history, tradition and cultural heritage? Future research should focus on the causality and consequences of cultural apathy. Respondents gave different views on the impact and expectations of the "European Capital of Culture" title. Vision was lacking in a couple of male respondents, who are not the best informed about the new title. By winning nominations, Novi Sad got the opportunity of presentations, affirmations, development of creativity and ideas, cultural training, which should be used in the best possible way. All citizens of good will need to be involved in that mission. Aggressive marketing could be essential for improving the perception of monitoring and understanding culture development in the 21st century. The prestigious title of the "European Capital of Culture" enhances the visible and intangible image of the city. It promotes the city at the European level. Positive effects are yet to be expected. As one of the respondents said, Novi Sad deserved it. This research will be the starting point for the future. In the near future, after the end of 2022, they will be able to see how much the title of European Capital of Culture has contributed to the improvement of the city's image.

The urban image is influenced by lifestyle, but also by residents. Some of them are so original and unrepeatable that they have become symbols of the city. These people, the symbols of the city, are often called 'urban legends'. The appearance of 'urban legends' of Novi Sad confirms that the urban image is changeable throughout history. Its intangible side is often unpredictable. In the distant future, the descendants of the inhabitants and new immigrants will have their own perception, which will be formed on the basis of the heritage that will be best preserved.

The COVID-19 pandemic hid people's faces. Also, it gave the city a cycling image. The most important thing is that it managed to bring out visible solidarity, understanding, and compassion from people. By bringing culture to the streets, the pandemic has added dynamism and variety to the city's image. Everything that could not be organized on the street, spread into the virtual world. The improved image of the city on the Internet has brought it closer to the so-called "distant world". Only two respondents out of fifteen gave comments in which no positive perception of the presence of the COVID-19 pandemic was stated. Therefore, it can be said that the hypothesis that negative phenomena, such as a pandemic, can result in a positive on urban image and visible cultural shifts has been confirmed. The work may represent a trace of the historical moment in which people lived with the pandemic. It can be valuable in some future image research when the pandemic is over.

The years to come should be used to make Novi Sad as a recognizable locality in Europe. The tourism traffic to be registered and the statistical parameters that speak of tourism revenues will be measures of success. The local population should culturally evolve in a positive sense and actively engage in the preservation of image and creation of a cultural future. The positive thing about all the problems is that, with the existence of good will, self-discipline, and self-awareness, they are solvable and it is possible to overcome them. The in-depth interview is a method that allows for examining the explanation of the attitudes and answers of the respondents. It limits the study to presenting only the most

extravagant parts, because otherwise it would be too extensive. In addition, the time that elapses before the publication of the paper brings certain changes, which the authors often regret that they will not be able to find in the paper.

**Author Contributions:** Conceptualization, T.L. and T.P.; methodology, I.B.; software, D.B.; validation, B.Đ; formal analysis, S.K.; investigation, M.C.; resources, M.B.Ž.; data curation, D.B.; writing—original draft preparation, T.L.; T.P. and S.K.; writing—review and editing, T.L.; visualization, B.Đ.; supervision, M.B.Ž.; project administration, M.C.; funding acquisition, I.B. All authors have read and agreed to the published version of the manuscript.

**Funding:** This research is funded by the Provincial Secretariat for Higher Education and Scientific Research of the Vojvodina Province, Serbia, as the part of the project numbered 142-451-2615/2021-01/2.

**Institutional Review Board Statement:** Institutional Review Board Statement is not required for this paper in Serbia. During the collection and storage of data, protection measures were established. Respondents were stressed that all data is collected exclusively for scientific purposes and that for other purposes it will not and cannot continue to be processed in a manner inconsistent with those purposes ("restriction on the purpose of processing"). As personal data is any data relating to a natural person whose identity has been determined or determined, directly or indirectly, especially on the basis of an identity mark, the respondents were not asked for data such as name and surname and identification number (JMBG), data about the nearest location and identifiers in electronic communication networks (e-mail, cookies), and for the features of his cultural and social identity, pseudonymization was performed, ie processing in a way that prevents assigning personal data to a certain person without using additional data.

**Informed Consent Statement:** Informed consent was obtained from all subjects involved in the study.

**Data Availability Statement:** Data available on request due to restrictions.

**Acknowledgments:** The authors are grateful to the reviewers, whose comments and criticisms have ensured the quality of the paper. Based on the reviewer's comments, the authors identified shortcomings and ambiguities and that the authors sincerely thank the reviewer for suggestions which will improve the future research.

**Conflicts of Interest:** The authors declare no conflict of interest.

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
