# Peer review of "Urban Image at the Time of the COVID-19 Pandemic, Case Study Novi Sad (Serbia)"

_societies, doi:10.3390/soc12020059_

Round 1
Reviewer 1 Report
The paper presents a local study, with a small number of interviewees (15 persons), in an attempt to have some representation, which generates serious limitations on research. The table shows the distribution of respondents by age, sex, origin and level of education, and the percentages resulting from the processing of these data are random, they can be radically changed by expanding the group of interviewees, or by choices made by operators.
Although in the sequence of the research method the authors refer to the way of conducting the interview (by phone, email, social networks, in physical format) the summary table does not contain this information regarding the work used and the choice of the interviewees. The section on the results obtained abounds with quotes from the interviewees, an aspect that can be restructured in the form of summary tables, indicating the code assigned to the respondent, or in the form of paraphrases in which the authors can summarize the information obtained from interview. A restructured format of this paper would highlight the originality of the paper, in which the authors will relate and interpret the information collected, which will lead to the final conclusions. The paper does not highlight the limitations of the research, which result from the small sample of the investigation, and does not show future options for improving the topic of research.
Regarding the list of references used by the authors, there is an unjustified loading of it, by including some secondary references, which can be replaced by the original sources, some of which are public and notorious. There are cases in which some references from the list of the paper become irrelevant, the context in which they are introduced in the text of the paper is given by the fact that the information comes from known public sources, which does not justify references to the papers, which can't be representative for this topic (eg. references 14,15,16 or ref. 52,53).
Author Response
Response to Reviewer 1 Comments
Point 1: The paper presents a local study, with a small number of interviewees (15 persons), in an attempt to have some representation, which generates serious limitations on research. The table shows the distribution of respondents by age, sex, origin and level of education, and the percentages resulting from the processing of these data are random, they can be radically changed by expanding the group of interviewees, or by choices made by operators.
Response 1: This reviewer's suggestion is understandable. However, the choice of the ‘in-depth interview’ method is simply that. No one method is perfect.
Point 2: Although in the sequence of the research method the authors refer to the way of conducting the interview (by phone, email, social networks, in physical format) the summary table does not contain this information regarding the work used and the choice of the interviewees.
Response 2: The authors do not consider that the presentation of this information is a function of the paper. At the same time, some respondents were communicated in several ways. Some respondents were contacted to clarify the written answers. Others asked themselves to supplement their statements.
Point 3: The section on the results obtained abounds with quotes from the interviewees, an aspect that can be restructured in the form of summary tables, indicating the code assigned to the respondent, or in the form of paraphrases in which the authors can summarize the information obtained from interview. A restructured format of this paper would highlight the originality of the paper, in which the authors will relate and interpret the information collected, which will lead to the final conclusions.
Response 3: The authors have already summarized thematically related responses. The authors do not have a consensus that additional summaries would help to interpret the results. The authors are grateful for this suggestion, because a careful review of the text revealed minor repetitions. They were corrected after that.
Point 4: The paper does not highlight the limitations of the research, which result from the small sample of the investigation, and does not show future options for improving the topic of research.
Response 4: The authors also thought about this suggestion. It is estimated that this would take the work away from the topic. The aim of the paper is not to argue about methodological advantages and disadvantages. Options for improving the research topic depend on a number of unpredictable factors, so the authors decided not to burden the work with that.
Point 5: Regarding the list of references used by the authors, there is an unjustified loading of it, by including some secondary references, which can be replaced by the original sources, some of which are public and notorious. There are cases in which some references from the list of the paper become irrelevant, the context in which they are introduced in the text of the paper is given by the fact that the information comes from known public sources, which does not justify references to the papers, which can't be representative for this topic (eg. references 14,15,16 or ref. 52,53).
Response 5: Thanks for this suggestion. Three references (14, 16, 52) have been replaced. No one was found better for the others. From the references, which are selected by the reviewer, generally known facts are taken.
Reference 14:
- Kovačić, S., Solarević, M., Pivac, T., Blešić, I., Lukić, T., Miljković, Đ., 2019. Slovak Cultural Heritage in Vojvodina (Serbia): Motives, Constraints for Visit and Sustainability Perspectives. Proceedings, International Scientific Symposium, New Trends in Geography, 70 Years Macedonian Geographical Society, Ohrid, Republic of North Macedonia, October 3 - 4, 2019, p.329. https://doi.org/10.37658/procgeo19329k
Replaced with:
- Bubalo-Živković, M., Đerčan, B. and Lukić, T., 2019. Changes in the number of Slovaks in Vojvodina in the last half century and the impact on the sustainability of Slovakia's architectural heritage. Zbornik radova Departmana za geografiju, turizam i hotelijerstvo, (48-1), pp.29-45.
Reference 16:
- Novaković, K. Odlike narodnog veza u Vojvodini. Etnologija. Rad muzeja Vojvodine 2004, 46. 151-184 UDC:746.3(497.113)
Replaced with:
- Ćurćić, S., Bubalo Živković, M., Đerčan, B. 2021. Geografija naselja, Zavod za udžbenike, Beograd, p 223., 978-86-17-20386-1
Reference 52:
- Simionescu, C.D., Rădoi, I. Facilitating Access to Cultural Heritage Through Cultural Mediation and Tourism. Case study: European Capital of Culture Timișoara 2021. Research Terminals in The Social Sciences, The Proceedings of CIL 2020: Ninth Edition of International Conference of Humanities and Social Sciences - Creativity, Imaginary, Language, Craiova, Romania, 22-23 May 2020. p.71.
Replaced with:
- Dragin, A.S., Zadel, Z., Mijatov, M.B., Stojanović, V., Jovanović, T., Lazić, L., Vasiljević, T.Z. and Milenković, N., 2021. Covid-19 Risk Management Perspectives of The European Capital of Culture: What Now?. Tourism in South East Europe..., 6, pp.195-213.
Reviewer 2 Report
line 37: what does "sociodemographic categories of the population" mean in this case?
line 248: write the acronym in full. readers don't know the meaning
line 285 to 289: repeated before
305 to 306: repeated before
316: somewhat radical statement. It may be relative to the respondents but it would possibly not represent the majority of those not born in the city.
566: has
580 to 581: is not well explained. The comment doesn't seem to be linked to the answers.
642 to 643: opinion of the author and not of the respondents. Maybe rephrase the sentence as a suggestion.
Author Response
Response to Reviewer 2 Comments
Point 1: line 37: what does "sociodemographic categories of the population" mean in this case?
Response 1: "Socio-demographic categories of the population" means to investigate the reactions of the population of Novi Sad, which is from different socio-demographic categories. The sentence has been corrected in the text.
Point 2: line 248: write the acronym in full. readers don't know the meaning
Response 2: Thanks for this suggestion. The acronym is explained in the text.
Point 3: line 285 to 289: repeated before
Response 3: Thank you very much for this suggestion. Corrected.
Point 4: 305 to 306: repeated before
Response 4: Thank you very much for this suggestion. Corrected.
Point 5: 316: somewhat radical statement. It may be relative to the respondents but it would possibly not represent the majority of those not born in the city.
Response 5: Yes, you're right. The paper adds that they are listed because of their unexpectedness and unusualness. Corrected.
Point 6: 566: has
Response 6: Thank you very much for this suggestion. Corrected.
Point 7: 580 to 581: is not well explained. The comment doesn't seem to be linked to the answers.
Response 7: The order in the sentences of the received answer has been rearranged, so that the comment is related to the answer.
Point 8: 642 to 643: opinion of the author and not of the respondents. Maybe rephrase the sentence as a suggestion.
Response 8: Thanks for this suggestion. You're right. Rephrased in the text.
Round 2
Reviewer 1 Report
The answer provided by the authors confirms that they have taken note of the reviewers' recommendations but maintain their chosen format for presenting this paper, and I respect this attitude.